# Molecular Aspects of Resistance to Immunotherapies—Advances in Understanding and Management of Diffuse Large B-Cell Lymphoma

**DOI:** 10.3390/ijms23031501

**Published:** 2022-01-28

**Authors:** Aleksandra Kusowska, Matylda Kubacz, Marta Krawczyk, Aleksander Slusarczyk, Magdalena Winiarska, Malgorzata Bobrowicz

**Affiliations:** 1Department of Immunology, Medical University of Warsaw, 02-097 Warsaw, Poland; aleksandra.kusowska@wum.edu.pl (A.K.); matyldakubacz@gmail.com (M.K.); marta.krawczyk@wum.edu.pl (M.K.); slusarczyk.aleksander@gmail.com (A.S.); magdalena.winiarska@wum.edu.pl (M.W.); 2Doctoral School, Medical University of Warsaw, 02-091 Warsaw, Poland; 3Laboratory of Immunology, Mossakowski Medical Research Institute, Polish Academy of Sciences, 02-106 Warsaw, Poland; 4Doctoral School of Translational Medicine, Centre of Postgraduate Medical Education, 01-813 Warsaw, Poland; 5Department of General, Oncological and Functional Urology, Medical University of Warsaw, 02-005 Warsaw, Poland

**Keywords:** diffuse large B-cell lymphoma, immunotherapy resistance, complement-dependent cytotoxicity, antibody-dependent cellular cytotoxicity

## Abstract

Despite the unquestionable success achieved by rituximab-based regimens in the management of diffuse large B-cell lymphoma (DLBCL), the high incidence of relapsed/refractory disease still remains a challenge. The widespread clinical use of chemo-immunotherapy demonstrated that it invariably leads to the induction of resistance; however, the molecular mechanisms underlying this phenomenon remain unclear. Rituximab-mediated therapeutic effect primarily relies on complement-dependent cytotoxicity and antibody-dependent cell cytotoxicity, and their outcome is often compromised following the development of resistance. Factors involved include inherent genetic characteristics and rituximab-induced changes in effectors cells, the role of ligand/receptor interactions between target and effector cells, and the tumor microenvironment. This review focuses on summarizing the emerging advances in the understanding of the molecular basis responsible for the resistance induced by various forms of immunotherapy used in DLBCL. We outline available models of resistance and delineate solutions that may improve the efficacy of standard therapeutic protocols, which might be essential for the rational design of novel therapeutic regimens.

## 1. Introduction

### 1.1. Diffuse Large B-Cell Lymphoma

Non-Hodgkin lymphoma (NHL) is the most frequent hematological neoplasm in the world [1] with more than 544,000 new NHL cases diagnosed in 2020 (2.8% of all cancer diagnoses) [2]. Of all the NHL subtypes, the most common is diffuse large B-cell lymphoma (DLBCL), accounting for approximately 40% of lymphoma cases. DLBCL is also one of the most aggressive subtypes; 5-year survival in elderly patients does not exceed 40% [3].

The most common first-line treatment for DLBCL is chemoimmunotherapy containing rituximab, the so-called R-CHOP regimen (rituximab, cyclophosphamide, doxorubicin, vincristine), which fails in 30–40% of patients. Although relapsed/refractory (r/r) patients receive second-line treatment or may undergo autologous stem cell transplantation, their prognosis remains poor [4]. As demonstrated by the SCHOLAR-1 trial, the objective response rate (ORR) of r/r patients treated with next-line therapy was 26%, and the median overall survival was 6.3 months [5]. The greatest challenge in DLBCL therapy is the genetic diversity of this neoplasm, i.e., the presence of mutations, translocations, and chromosomal aberrations. In the face of this heterogeneity, investigation of novel common antigens, which can be targeted by immunotherapy and identified by a universal approach to eliminate malignant cells regardless of their molecular pathogenesis, is constantly pursued [6].

Recently, r/r patients have gained the access to adoptive therapy using genetically engineered T cells with chimeric antigen receptor (CAR-T cells) targeting DLBCL-expressed surface antigens. Particularly promising results of this treatment strategy have been observed in the case of CD19-CARs. Three of them, i.e., axicabtagene ciloleucel (Yescarta, Kite), lisocabtagene maraleucel (Breyanzi, Bristol Myers Squibb), and tisagenlecleucel (Kymriah, Novartis) are licensed to treat DLBCL patients [7].

However, years of employment of rituximab as a core of first-line treatment and recent observations on CAR-T cell therapy clearly demonstrate that the use of immunotherapy invariably leads to the induction of resistance [8]. A deep understanding of the molecular mechanisms underlying this phenomenon is essential for the rational design of novel therapeutic regimens. Therefore, this review aims at summarizing the mechanisms of resistance induced by various forms of immunotherapy used in DLBCL (Figure 1).

### 1.2. Rituximab (RTX)

Targeting CD20 antigen using rituximab, a monoclonal antibody (mAb) approved by the FDA in 1997, remains the most widely used form of immunotherapy in oncology. CD20 is a membrane protein found on the surface of normal and neoplastic B lymphocytes, but not on hematopoietic progenitors, which makes it a suitable target for mAbs [9]. In clinical trials, rituximab combined with chemotherapy significantly improved progression-free survival (PFS) and overall survival (OS) rates in B-cell-derived malignancies such as chronic lymphocytic leukemia (CLL), follicular lymphoma (FL), and DLBCL [10]. Experience with rituximab provided proof of concept that mAbs could be used in cancer treatment and the combination of rituximab with chemotherapy, i.e., the R-CHOP regimen has remained the first-line of treatment for DLBCL patients for years.

### 1.3. Mechanisms of Rituximab Cytotoxicity

The cytotoxic effect of rituximab is exerted through several mechanisms, relying on the activation of anti-tumor immune response (reviewed in [11,12]). These include complement-dependent cytotoxicity (CDC), antibody-dependent cellular cytotoxicity (ADCC), antibody-dependent cellular phagocytosis (ADCP), and direct apoptosis induction [13]. Additionally, rituximab sensitizes non-Hodgkin lymphoma cells to chemotherapy [14,15]. Rituximab hampers the secretion of interleukin (IL)-10, which is produced by NHL cells as a protective factor. IL-10 induces STAT3 phosphorylation and in consequence the expression of an anti-apoptotic protein Bcl-2, which leads to the resistance to chemotherapeutics [14,15]. Moreover, by upregulating Raf-1 kinase inhibitor protein (RKIP), rituximab decreases the signaling of the ERK1/2 and NF-kappa B pathways, leading to downregulation of Bcl-xL [15]. Furthermore, in NHL cell lines, rituximab inhibits the overexpressed transcription repressor Yin Yang 1 (YY1), which suppresses Fas and DR5 expression, leading to sensitization to Fas ligand and tumor necrosis factor (TNF)-related apoptosis-inducing ligand (TRAIL)-induced apoptosis [16].

The results of preclinical studies in vitro and in vivo in animal models as well as observations from the clinical settings suggest that all the above-mentioned mechanisms contribute to the overall efficacy of rituximab. However, it seems that the influence of single effector mechanisms of rituximab may vary depending on the tumor type.

## 2. Resistance to Antibody-Dependent Cellular Cytotoxicity

It has been proposed by several studies that ADCC is the leading mechanism of action of rituximab in lymphomas [17,18,19,20]. ADCC relies on the recognition of Fc fragment of the antibody that opsonizes the target cell by the Fc receptor (FcR) present on the surface of the effector cell, which in turn undergoes degranulation. It has been demonstrated that natural killer (NK) cells are the major effectors of ADCC [21]. This effect is predominantly exerted by a subpopulation of CD56^dim^ NK cells with higher levels of FcγRIII receptor (CD16) [22]. Due to their ability to mount a rapid immune response and no need for priming, they play a crucial role in initiating anti-tumor response [23]. By now, there is a strong body of evidence describing various factors shaping the primary resistance and warranting the worst prognosis. Thus far, neither in vitro nor in vivo models studying the development of resistance through rituximab-induced ADCC have been generated.

### 2.1. Mechanisms of Primary Resistance to ADCC

#### 2.1.1. Effector Cell Characteristics Influencing ADCC Efficacy

##### Genetic Polymorphism in FcγRIII Receptor

The efficiency of NK cell-mediated ADCC is highly influenced by the ability of FcγRIII receptor to recognize Fc portion of immunoglobulins [24]. Polymorphisms in the gene encoding FcγRIIIa (CD16a) determine response rates to R-CHOP treatment. The polymorphism at nucleotide position 559 (G or T) leads to the synthesis of valine (V) or phenylalanine (F), respectively. This in turn results in three different genotypes: homozygous *FcγRIIIa-158 F/F* or *FcγRIIIa-158 V/V*, and heterozygous *FcγRIIIa-158 V/F*, each characterized by diverse binding affinity to Fc portion of mAbs, and thus the variable rate of NK-cell cytotoxicity [25]. Among them, *V/V* genotype was described as high-affinity, whereas *F/F* as a low-affinity variant [26]. Several studies demonstrated the advantage of *V/V* over *F/F* genotype in association with the response to rituximab and R-CHOP regimen [27,28], but also with PFS and OS in large cohorts of DLBCL patients following R-CHOP treatment [29,30,31], especially in germinal center-derived (GCB) subset of DLBCL patients [32]. On the contrary, some studies have questioned the importance of *FcγRIIIa* polymorphism in mediating effective ADCC response to R-CHOP [33] or its role in predicting prognosis in DLBCL [34], which suggests the existence of other factors influencing the efficiency of ADCC in tumor eradication.

##### Low CD16 Expression

One of the reasons for the impaired ADCC activity in DLBCL patients might be an inherently low expression of CD16 on effector cells. A study on 36 newly diagnosed DLBCL patients presented markedly decreased protein levels of CD16 and CD137, a co-stimulatory molecule expressed on activated NK cells, as compared to healthy controls. This resulted in impaired degranulation in the presence of RTX, which could not be rescued by stimulation with IL-2 [35]. A comprehensive analysis of phenotypic and functional characteristics of NK cells at DLBCL diagnosis followed by RTX treatment revealed that the initial high level of NK CD16-positive cells including CD56^dim^ population decreased over the course of therapy, whereas the expression of activating receptors (CD16, NKG2D) was diminished [36].

##### KIR/HLA Interactions

ADCC function of NK cells is strongly influenced by the interactions between inhibitory killer cell Ig–like receptors (KIRs) and human leukocyte antigen (HLA) class I on target cells [37]. The binding of some KIR ligands to KIRs hampers NK-cell degranulation [38]. A study on NK cells isolated from the blood of healthy donors revealed that the interaction of KIR2DL1/KIR2DL2/KIR2DL3 with HLA-C and KIR3DL1 with HLA-Bw4 on EBV-B cell line significantly reduces rituximab-mediated degranulation of educated NK cells and ADCC efficacy [39]. Further, investigation of clinical response to rituximab in 74 NHL patients demonstrated that low KIR ligand number and high frequency of KIR-positive NK cells, which are notably efficient in driving RTX-mediated ADCC, correlated with improved complete response (CR) and prognosis. Thus, it has been postulated that low KIR ligand number and peripheral KIR-negative NK cells are good prognostic markers of response to RTX-based regimens [39].

However, to the best of our knowledge, it has not been demonstrated thus far whether lymphoma cells modulate the expression of KIR ligands on their surface or induce changes in the expression of KIRs on the surface of effector cells following rituximab treatment.

#### 2.1.2. The Influence of the Microenvironment on ADCC Efficacy

Recent findings indicate that the presence of other immune cells may affect ADCC potency. Depletion of NK cells from peripheral blood mononuclear cells (PBMCs) entirely abrogated ADCC, whereas removal of monocytes increased ADCC efficacy [40]. Further, pre-incubation of rituximab-coated lymphoma cells with THP-1 monocyte cell line resulted in the suppression of NK cell-mediated ADCC [40] since monocytes were found to remove rituximab from the cell surface of B-cells [41]. The phenomenon of monocyte-mediated shaving of RTX/CD20 complexes was found to rely on serine protease activity since protease inhibitors can partially suppress RTX removal from the cell surface [42].

Monocytes and neutrophils can produce NOX2-derived reactive oxygen species (ROS) known to play immunomodulatory functions [43]. NK cells were found to be particularly sensitive to oxidative stress induced by hydrogen peroxide (H_2_O_2_), which markedly decreased NK-cell viability and significantly impaired the ability to mediate ADCC. The reduced anti-tumor potential of NK cells was correlated with intrinsically low expression of peroxiredoxin-1 (PRDX1), whereas its overexpression improved NK-cell function [44].

Anti-CD20 mAbs promote the release of ROS from neutrophils in CLL patient samples [45] and lymphoma cell lines [46]. Rituximab and ofatumumab were shown to trigger monocytes to generate ROS in a NOX-2 mediated pathway by attaching to Fc portion of anti-CD20 mAb [47]. In turn, NK-cell mediated ADCC against primary CLL cells was markedly reduced in the presence of monocytes and only partial rescue of ADCC function occurred in presence of anti-oxidants suggesting the involvement of more complex mechanisms associated with monocyte-mediated ADCC inhibition [47].

#### 2.1.3. Characteristics of Malignant B-Cells Contributing to Impaired ADCC

It is widely accepted that tumor cells develop strategies to avoid recognition and eradication by the immune system. This is also true for DLBCL cells [48,49]. Data from The Leukemia and Lymphoma Molecular Profiling Project that analyzed gene expression in 240 DLBCL patient samples demonstrated reduced surface expression of MHC and CD58 molecules in DLBCL, which inhibits T-cell and NK-cell infiltration [50,51]. In addition, aberrant expression of anti-apoptotic and pro-apoptotic proteins in DLBCL reduces NK-cell cytotoxicity [52]. For instance, Bcl-2 protein is aberrantly overexpressed in 20% of DLBCL cases due to chromosomal translocation t(14;18) that juxtaposes *Bcl-2* to the immunoglobulin heavy chain gene enhancers resulting in survival promotion [53]. Likewise, inactivating mutations in pro-apoptotic Noxa in DLBCL [54] contribute to decreased efficacy of NK-cell cytotoxicity [52]. Moreover, NHL cells express high levels of serpin B9, a serine protease inhibitor known to mitigate the tumoricidal activity of NK cells by blocking degranulation [55].

### 2.2. Studies on Acquired Resistance

#### 2.2.1. ADCC-Resistance Models to Study Acquired Rituximab Resistance

The availability of in vitro models representing ADCC resistance in B-cell lymphoma is scarce. Czuczman et al. induced rituximab resistance (RR) in Raji (Burkitt lymphoma) and SU-DHL-4 (DLBCL) [56], whereas Jazirehi et al. in Daudi and Ramos (Burkitt lymphoma) cell lines [57] by incubations with escalating doses of RTX +/− human serum. Although these cell lines exhibit resistance to cell death mediated by RTX not only in CDC but also in ADCC mechanism, they were not generated by co-culture with NK cells and may not reflect the exact genetic and phenotypic changes that might have occurred upon this process. A study on RTX-resistant cell lines (RRCLs) demonstrated that point mutations in complementarity-determining regions (CDRs) of RTX induced enhanced ADCC and restored apoptosis in RR lymphoma cells as compared to wild-type RTX [58]. Therefore, enhancing the avidity of already existing mAbs might rescue or potentiate ADCC capability in r/r lymphoma.

Interesting data come from the study of Aldeghaither et al., who successfully induced ADCC resistance in squamous carcinoma cell line (A431) highly expressing epidermal growth factor receptor (EGFR) by co-culture with NK92-CD16V cells and incubation with anti-EGFR mAb–cetuximab [59]. This study proposed that ADCC resistance was primarily attributed to the impaired NK-cell activation caused by defects in immune synapse formation and cell-to-cell conjugation, rather than by a decrease in target antigen expression [59]. Similar observations were reported in acute lymphocytic leukemia (ALL) murine models, where the failure of treatment with T-cell receptor mimic monoclonal antibody (TCRm mAb) was not associated with target antigen loss or downregulation [60] but with a kinetic escape mechanism [61].

#### 2.2.2. RTX-Mediated Changes in NK Cells

Rituximab causes phenotypical and functional alterations in NK cells from healthy donors [62]. Pre-incubation of non-malignant PBMCs with RTX led to NK-cell degranulation, downregulation of CD16, and depletion of healthy B cells, whereas in CD56^dim^ cells, the expression of NK-cell activation markers, i.e., CD69 and CD137 was significantly increased, as compared to untreated cells. Another study demonstrated that CD16 downregulation is partially caused by its internalization upon interaction with RTX- or ofatumumab-coated B cells [63]. Defects in the cytotoxic potential of NK cells were found to correlate with some cross-inhibited NK-activating receptors, i.e., NKG2D, NKp46, 2B4, DNAM-1, which rendered primary NK cells from CLL patients hyporesponsive [63]. Further characterization of hyporesponsive NK cells revealed defects in their ability to phosphorylate some signaling molecules essential for NK-cell mediated cytotoxicity, i.e., SLP-76, PLCγ2, and Vav1 and consequently to prolonged recruitment of phosphatase SHP-1, which promoted desensitization of activated receptors [63].

Altogether, impaired basal ADCC activity might be associated with treatment failure in the future, whereas the negative impact of therapy-induced alterations of NK cells may partially explain the development of ADCC resistance.

### 2.3. Solutions to Improve ADCC Efficacy

#### 2.3.1. Fc Receptor Engineering

Modification of Fc fragment of mAb to improve its binding affinity to FcγRIIIa proved to stimulate enhanced ADCC and CDC against lymphoma and leukemia cells as compared to unmodified RTX [64]. Fc-engineering strategy has been also employed in the generation of anti-CD20 mAb– obinutuzumab (OBI) currently registered for the treatment of CLL and FL [65]. OBI demonstrated markedly enhanced binding affinity to FcγRIIIa and dramatically improved potency of ADCC in comparison to RTX. Importantly, OBI-mediated activation of NK cells was reported to occur irrespective of KIR/HLA interactions [66,67]. Although OBI (Gazyva) combined with CHOP (G-CHOP) was not superior over R-CHOP in clinical trials involving previously untreated DLBCL patients (NCT01659099, NCT01287741), some benefit of G-CHOP in terms of PFS was associated with GCB subtype of DLBCL [68,69]. Nevertheless, the advantage of OBI over RTX has been reported in a large cohort of 1202 advanced-stage FL patients [70] and in 781 previously untreated CLL patients [71].

Another Fc-engineered anti-CD20 mAb–ocaratuzumab (AME-133v, currently in clinical trials–NCT01858181), showed more potent ADCC function compared to RTX in several clinical trials involving r/r FL patients [72,73], also in association with unfavorable *FcγRIIIa-F* genotype [74,75]. An improved binding affinity to FcγRIIIa and potentiation of NK cell-mediated ADCC was as well achieved by the introduction of S239D and I332E amino acid substitutions to generate anti-CD19 mAb–tafasitamab (MOR208) [76], which was highly effective in the treatment of r/r DLBCL and r/r FL patients [77].

#### 2.3.2. Supplementation with Cytokines

Many efforts have been taken to augment NK-cell anti-tumor responses. Stimulation of NK cells with ALT-803, a super agonist of IL-15, a cytokine stimulating proliferation of these cells, resulted in enhanced ADCC against B-cell lymphoma in vitro and in vivo [78]. It has been demonstrated that supplementation with vitamin D3 significantly enhanced the efficacy of NK cell-mediated ADCC in elderly DLBCL patients and that rituximab-treated individuals suffering from vitamin D3 deficiency have worse survival [79]. Therefore, it has been suggested that vitamin D3 positively influences rituximab-mediated ADCC, and attempts to determine optimal serum levels of 25-OH-D3 for maximum NK-cell activity have been undertaken [80]. Unfortunately, a mechanism responsible for improving ADCC by vitamin D3 has not been published thus far.

Although molecular mechanisms underlying resistance to ADCC have not yet been thoroughly investigated specifically in DLBCL, available literature data suggest some factors involved. The existing ADCC in vitro and in vivo models propose that alterations in signaling pathways, defective NK-cell activation, and kinetics of cancer cell growth rather than target antigen loss might potentially play a crucial role in the development of resistance.

## 3. Resistance to Complement-Dependent Cytotoxicity

CDC is mediated by the classical pathway of the complement system upon binding of C1 complex to RTX-opsonized cells. This triggers activation of the complement cascade which results in the formation of the membrane attack complex (MAC) in the target cell membrane leading to a loss of membrane integrity and cell lysis [81].

Although ADCC is suggested as the main mechanism of action of RTX in clinics [15,16,17], the importance of complement is also postulated [82]. CDC seems to play a predominant role in mediating rituximab-induced cell death in CLL [83] due to high leukocytosis and therefore high accessibility of tumor cells to complement complex, which is reflected by the rapid exhaustion of complement and C3b(i) deposition [84]. In addition, side effects after infusion can be attributed to complement cascade activation [83]. Moreover, an extensive body of research demonstrated that CLL patients refractory to rituximab are often characterized with complement deficiencies, correlates with advanced disease, and shorter survival time [85,86,87]. In line with these findings, administration of individual complement components restores CDC efficacy in complement-deficient patients [88], and concurrent administration of rituximab and fresh frozen plasma remarkably enhanced the therapeutic activity of RTX [89]. In addition, CLL cells resistant to rituximab have significantly higher expression of complement regulatory proteins (CRPs) than the sensitive cells [90]. In CLL, primary resistance to CDC is mediated also by the presence of soluble protective protein complement factor H (CFH) that prevents the formation and deposition of additional C3b and propagation of the complement cascade. Additionally, inhibiting or inactivating CFH with monoclonal antibody permits activation of complement leading to cell lysis and death in resistant CLL cells [91].

In DLBCL patients, given the scarce presence of malignant cells in the circulation and therefore lower accessibility of tumor cells to complement complex, it would seem that the main benefit of incorporating rituximab into CHOP relies on its ability to chemosensitize malignant cells. However, it has been demonstrated in a disseminated lymphoma tumor mouse model that C1q-deficient mice exhibited a defective response to RTX, while depletion of either NK cells or granulocytes did not hamper rituximab efficacy [92]. In a human-CD20 transgenic mouse model, Gong et al. showed that circulating B cells were depleted through ADCC, while B cells in the marginal zone of lymph nodes were lysed primarily by CDC [93].

Some studies show an essential role of complement in determining the clinical efficacy of rituximab also in DLBCL, as high expression of CD59 correlates with poor OS and PFS [94]. CD59 is one of CRPs that blocks the formation of MAC, which is the last step of complement activation and thus prevents cell lysis [95]. In line with these findings, the levels of miR-224, a negative regulator of CD59 expression, are significantly lower in DLBCL patients with poor OS and PFS [94].

The engagement of CDC in the clinical success of rituximab in NHL is a matter of controversy, as it has been suggested that complement activation can be detrimental to antibody immunotherapy in murine lymphoma models. It has been demonstrated that breakdown products of the complement component C3 block rituximab activity and render NK cells less active in ADCC [96]. In line with that, in vitro studies demonstrated that NK-cell activation and RTX-induced ADCC are inhibited by the C3b component of complement [97]. Moreover, polymorphisms in *C1q* gene allele were recently shown as important factors that possibly influence the success of therapy with rituximab in NHL. It has been demonstrated that FL patients homozygous for the *C1qA[276]G* allele, which most probably correlates with higher serum levels of C1q and thus higher complement activity, have a significantly shorter time to progression than A carriers [98]. An analysis of DLBCL patients demonstrated that homozygotes for the *C1qA[276]G* allele had not only shorter PFS but also decreased OS and CR rates [99]. Nevertheless, the role of complement in determining the efficacy of rituximab in NHL remains an open question and requires further investigation.

### 3.1. The Role of the Cell Membrane Composition in Determining Primary Resistance to Rituximab

The binding of rituximab to CD20 induces the translocation of the antibody–antigen complex to the lipid rafts [100], highly dynamic domains in the cell membrane rich in cholesterol and sphingolipids that serve as docking points for signal transduction [101]. It is recognized that the composition of the lipid raft domains may play an important role in determining resistance to rituximab [102]. In CLL patients, resistance to rituximab was shown to be associated with aberrant recruitment of Csk-binding protein, leading to a lack of sphingomyelin and ganglioside M1 at the outer leaflet of the plasma membrane of tumor cells [103]. Furthermore, low ganglioside M1 levels in CLL and MCL patients resulted in a lack of sensitivity to RTX-induced CDC in vitro [104]. Moreover, a study on primary CLL samples demonstrated that resistance to CDC is strongly associated with cell surface α2-6 hypersialylation induced by increased activity of α2-6 sialyl transferase and that the removal of terminal sialic acids enhances the activity of rituximab-induced CDC [83]. Importantly, hypersialylation that leads to binding of factor H is a known factor protecting erythrocytes from complement-induced lysis [105]. Observations from MCL primary samples and cell lines show that malignant B cells produce aberrant lipid rafts leading to impaired signaling transduction as a way to avoid apoptosis [106]. Similar findings have been published for DLBCL, where the altered organization of lipid rafts led to the resistance to conventional chemotherapy [107]. Additionally, the presence of plasma membrane cholesterol was found to promote conformational changes in CD20 antigen, which are critical for the effective binding of anti-CD20 mAbs and thus potent RTX-mediated CDC [108]. Therefore, it is highly probable that membrane composition influences response to rituximab also in this type of lymphoma.

### 3.2. Rituximab-Resistant Cell Lines as a Tool to Study the Mechanisms of Acquired Resistance to CDC

Already 10 years after rituximab introduction to the clinic, two large studies on r/r DLBCL patients clearly demonstrated that rituximab used as the first-line treatment leads to substantial alterations in the biology of malignant cells and makes them resistant to subsequent salvage regimens [109,110]. Early relapses following rituximab-containing first-line therapy have a poor prognosis [109] and prior exposure to rituximab was demonstrated to be an independent adverse prognostic factor for both PFS and OS in r/r DLBCL [110].

To better understand the mechanisms of resistance, cell line models have been generated by Czuczman et al. [56], Barth et al. [111], Takei et al. [112], and Jazirehi et al. [57]. They were generated by repeated exposure to elevated concentrations of rituximab or rituximab plus human serum as a source of complement, thus they mimic the development of resistance to CDC-dependent cell death. The generated cell lines have become important models for studying the molecular mechanisms of resistance induced by prolonged exposition to rituximab.

First of all, they enabled to determine that CD20 downregulation occurs following exposure to rituximab is regulated on both transcriptional and posttranslational levels [56,113]. Second, upregulation of complement inhibitory proteins (CD46, CD55, CD59) in RRCLs has been reported [56]. This corresponds with the data obtained from in vitro functional analyses on primary malignant cells from both NHL [114] and CLL [90], where the levels of CRPs (mainly CD55) were suggested as markers for clinical response to rituximab treatment.

### 3.3. CD20 Downregulation in Acquired Resistance to Rituximab

Resistance to rituximab and R-CHOP results as a consequence of molecular alterations of both target malignant cells, effector immune cells, and the influence of tumor microenvironment. One of the most immediate causes of resistance to anti-CD20 mAbs are decreased CD20 levels [13,115]. Yet, the importance of CD20 downregulation in real-world settings has not been sufficiently documented and it is based on case reports [84,115,116,117,118,119]. Even if the response rate at relapse to rituximab in prior responders is below 50%, these patients are not routinely re-biopsied [117].

CD20 levels have been shown by several groups to correlate with the efficacy of rituximab (reviewed in [13]). Experimental in vitro studies using cell clones with divergent CD20 expression demonstrated a direct correlation between the number of CD20 molecules on the cell surface and CDC efficacy [120]. Intriguingly, no such correlation has been observed in ADCC efficacy [120]. However, our observations on increasing CD20 levels with small-molecule inhibitors showed improved efficacy of rituximab in SCID mice, where NK-cell mediated ADCC is one of the leading effector mechanisms [121].

The biological function of CD20 is still unclear despite years of extensive research. CD20 is described as a component of a store-operated calcium entry pathway activated by B-cell receptor (BCR) [122]. Until now, the existing body of evidence has suggested that CD20 loss in humans and in murine models leads to a relatively mild phenotype (reviewed in [13]) with no alterations in B-cell survival nor in proliferation rate [123]. Therefore, as CD20 is not indispensable for cell survival, it is not surprising that tumor cells downregulate this target molecule. The mechanisms leading to CD20 downregulation has been reported by several groups based both on clinical observations and preclinical in vitro studies (reviewed in [13]) and include all steps of protein synthesis–transcription [124,125,126], translation as well as post-transcriptional and post-translational modifications [121] and transport of CD20 to the cell membrane [108,127,128]. Moreover, the presence of a soluble form of CD20 protein reported in CLL patients [129] and CD20 secreted in exosomes in patients suffering from aggressive NHLs [130], which compete with CD20 for binding to rituximab, contributes to the reduction of the effectiveness of mAbs. There are also reports on trogocytosis—“shaving” and endocytosis of the antigen by phagocytic cells [41], but it has to be kept in mind that this mechanism has been proven only in in vitro settings.

The mechanisms of CD20 regulation and potential compounds that may be combined with rituximab in order to increase its efficacy by inducing CD20 upregulation are elegantly reviewed in [13,131]. A genome-wide study by de Jong et al. provides data on rituximab compatible drug-target combinations for DLBCL [132].

### 3.4. Hexamerization-Inducing Mutations as a Way to Improve CDC Efficacy

Recently, it has been demonstrated that Fc fragments of IgG antibodies form ordered Ab-hexamers following antigen binding [133]. The introduction of hexamerization-inducing mutations (E345 or E430) in *IgG1* gene strongly augmented the ability of some Abs to induce CDC and ADCC against lymphoma cell lines and primary CLL samples [134]. Since both mutations were found to play a stimulatory role in inducing CDC, the presence of the conserved E residue might be responsible for the suppression of excessive complement activation [134].

## 4. Mechanisms of Chemoresistance following Rituximab-Treatment

Given the mentioned poor prognosis following the first-line treatment with rituximab in relapsed patients, the mechanisms of chemoresistance have also been thoroughly studied in the models of RRCLs. All the RRCLs are resistant to cisplatin, doxorubicin, paclitaxel, and vincristine due to decreased expression of the Bcl-2 family proteins, i.e., Bax, Bak, Bcl-2, and increased expression of Bcl-xL and Mcl-1 [135,136]. Therefore, it can be concluded that while a limited exposure of lymphoma cells to rituximab may lead to a rise in the number of pro-apoptotic proteins sensitizing malignant cells to chemotherapy, repeated exposure leads to the induction of an apoptosis-resistant phenotype [135]. Prolonged exposure of cell lines to rituximab leads also to downregulation of ICAM-1, which impairs cellular adhesion and aggregation [137]. Importantly, ICAM-1 was demonstrated to be a marker of positive response in NHL rituximab-treated patients. It has been postulated that ICAM-1 downregulation in RRCLs may contribute to the development of chemoresistance, as cell clustering upon rituximab-treatment increases the efficacy of chemotherapeutics [137].

RRCLs are also characterized by profound metabolic changes. It has been demonstrated that the acquirement of rituximab resistance is associated with deregulation in glucose metabolism caused by overexpression of hexokinase II (HKII), which performs dual functions of promoting glycolysis as well as inhibiting mitochondrial-mediated apoptosis [138]. These data have been next confirmed in gene expression profiling datasets from R-CHOP treated patients and higher levels of HKII are considered to correlate with poor PFS and OS in DLBCL [138].

## 5. The Role of Tumor Microenvironment in Promoting Resistance to Rituximab-Containing Regimens

Recently, more attention has been given to the role of the tumor environment in promoting tumor progression and influencing the clinical response to treatment [139]. In NHL patients where only a minor part of malignant cells is found in the circulation, the role of the microenvironment is particularly important [140]. For instance, mesenchymal stromal cells (MSCs) decrease the efficacy of anti-CD20 treatment by decreasing CD20 expression [141]. Moreover, adhesion to stromal cells protects malignant B cells from apoptosis induced by chemotherapeutics as well as from rituximab-induced apoptosis [142]. Blocking the interaction between tumor and stromal cells with natalizumab, which targets VLA-4 (integrin alfa-4-beta-1/CD49d), partially overcomes this protection [142].

Although bone marrow infiltration is not common in DLBCL, the studies of this milieu in the context of therapy efficacy need to be developed as bone marrow-derived MSCs have been shown to promote the growth and primary resistance of DLBCL cell lines to rituximab by secreting IL-6 and elevating IL-17A levels in in vitro settings [143]. Increased levels of IL-6 have been demonstrated in the immunohistochemistry of tumor tissues from DLBCL patients [143]. Importantly, rituximab itself may induce tumor cells to produce IL-6 and consequently induce T-regulatory cells (Tregs) to secrete IL-17, which leads to tumor growth and rituximab resistance in DLBCL patients [144]. Therefore, it has been postulated that rituximab could be a “double-edged sword” and induce resistance and poor prognosis in relapsed DLBCL patients [144].

Conversely, rituximab has been shown to promote long-time response due to its vaccinal effect as demonstrated in EL4-huCD20 murine models of lymphoma that were resistant to tumor expansion at rechallenge with RTX [145]. Further studies have shown that rituximab administration reverses the expansion of immunosuppressive Tregs and increases the number of pro-inflammatory T-helper 1 (Th1) cells. Rituximab treatment led to an increase in IL-12, increased numbers of activated myeloid dendritic cells, and CD4-positive effector memory T cells in treated animals [146].

As DLBCL cells predominantly reside in lymph nodes, the stromal factors implicated in the organization of extracellular matrix (ECM) and the cells of innate immunity, especially macrophages, dendritic cells (DCs), and NK cells are involved in the physiopathology and drug response of DLBCL [147]. A study by Lenz et al. described a “stromal” signature characterized by high deposition of ECM proteins, e.g., fibronectin, secrete protein acid rich in cysteine (SPARC), and vast infiltration of myelo-monocytic cells [148]. Such signature leads to prolonged survival following R-CHOP suggesting an adjuvant role of macrophages in promoting rituximab cytotoxicity. In contrast, the presence of an immunosuppressive population of M2-like macrophages was shown to correlate with shorter survival in R-CHOP-treated cohorts [149]. Therefore, therapeutic strategies aiming at promoting the differentiation to M1 macrophages may improve prognosis in DLBCL.

### Targeting Immune Microenvironment as a Way to Improve Rituximab Efficacy

In light of the above, modifying the tumor environment with immunomodulatory drugs seems a rational solution. One of such drugs is lenalidomide, currently approved by FDA for MCL treatment [150]. In an international phase II trial of r/r B-NHL patients, lenalidomide as a single agent had ORR of 35% [151]. Combination with rituximab seems to be synergistic and highly effective, as 90% of previously untreated advanced-stage low-grade NHL patients reported a CR [152]. This effect may be due to lenalidomide-mediated activation of DCs and increased recruitment of NK to the tumor site [153]. Multiple phase I/II trials testing combinations of lenalidomide with R-CHOP in hope of improving long-term outcomes are underway.

## 6. Other Treatment Modalities as Possible Solutions for Relapsed/Refractory Patients

### 6.1. Novel Targets for mAbs

Despite the success of rituximab as a safe and efficient therapeutic option, the incidence of resistance encouraged searching for other molecular targets for immunotherapies in the management of DLBCL (Figure 2). These novel targets, including CD19, CD22, CD40, CD80, and CD79b, have been comprehensively reviewed in [7,154,155,156]. The most successful mAbs in recent times are the ones targeting CD19–tafasitamab registered recently in a combination with lenalidomide and antibody–drug conjugates (ADCs), i.e., loncastuximab tesirisine-lpyl and an anti-CD79b ADC–polatuzumab vedotin. The approval of polatuzumab vedotin in combination with bendamustine and rituximab in r/r DLBCL was preceded by a clinical trial that for the first time after several years demonstrated the benefit in OS to an experimental arm of the study [154]. Besides the above-mentioned targets, an attractive alternative is the use of CD37-directed mAbs, which are currently in clinical trials and we have recently reviewed them in [157].

### 6.2. Bispecific Antibodies

A way to improve the efficacy of treatment response is to employ dual antigen targeting with bispecific Abs (bsAbs) to stimulate a more potent anti-tumor response, increase binding specificity and overcome the low efficacy of single-epitope targeting. Moreover, the inclusion of single-chain variable fragment (scFv) of an anti-CD16 Ab into bsAb constructs has been employed to enhance CD16-positive effector cell recruitment [158]. In addition, utilizing the cytotoxic potential of T cells in synergy with conventional mechanisms of mAb-mediated cancer cell death has been recently exploited in the generation of bispecific T-cell engagers (BiTEs). Examples of bsAbs and BiTEs, their mode of action, and efficacy are summarized in Table 1.

### 6.3. Antibody–Drug Conjugates

Antibody–drug conjugates are composed of specific mAbs linked to cytotoxic agents, the so-called free payload, that exert their cytotoxic action upon internalization [171]. Since the cytotoxic agents are released at the tumor site the likelihood of systemic exposure and toxicity is therefore significantly lower. Additional advantage comes from the immunogenic cell death, as the release of damage-associated molecular patterns (DAMPs) by the tumor cells is known to stimulate antigen-presenting cells (APCs) [172].

Several specific markers of B cells that may be used as targets for ADCs have been identified in in vitro studies, i.e., CD19, CD20, CD21, CD22, CD72, CD79b, and CD180 [173]. By now, a few ADCs have been registered as treatment options for B-cell malignancies, i.e., brentuximab vedotin (CD30-positive Hodgkin lymphoma (HL); anaplastic large-cell lymphoma (ALCL) [174], inotuzumab ozogamicin (r/r B-cell precursor ALL) [175], polatuzumab vedotin (r/r DLBCL) [176] and gemtuzumab ozogamicin (for CD33-positive acute myeloid leukemia (AML)). ADCs against various lymphoma types with the highest clinical potential are summarized in Table 2.

Based on in vitro studies and xenograft models of repeated exposure to consecutively increasing concentrations of ADCs, several leading cellular mechanisms, by which the acquired resistance to ADCs occurs were described, each elegantly reviewed by Loganzo et al. [177]. The first resistance mechanism is antigen downregulation or mutation [178]. Moreover, the development of resistance was found to be correlated with the overexpression of ATP-binding cassette (ABC) transporters such as MDR1, MRP1, and BCRP, which play a significant role in the cellular efflux of multiple antineoplastic agents [179] such as MMAE, DM1, and ozogamicin [178]. Brentuximab vedotin-refractory HL, patients were found to have upregulation of the multidrug resistance *MDR1* gene [180]. The expression of ABC transporters is increased also in DLBCL cells, where it correlates with a worse response to treatment [181].

Several attempts are undertaken to enhance the therapeutic potential of ADCs [182]. Alterations within the antibody (to obtain sufficient antigen affinity and specificity), linker or the payload (DNA-interactive payload classes or microtubule inhibitors, e.g., calicheamicins, auristatins, and maytansinoids) increase ADC efficacy [182,183]. Another approach is including ADCs in multi-drug schemes, e.g., testing inotuzumab ozogamicin with R-CHOP for the treatment of DLBCL patients unfit for anthracycline has been evaluated in NCT01679119 study [184].

**Table 2 ijms-23-01501-t002:** Antibody–drug conjugates in clinical trials.

Name	Target Antigen	Cytotoxic Payload and its Mechanism of Action	Possible Application of ADC	Clinical Efficacy of ADC
Polatuzumab vedotin (PV)	CD79b	Monomethyl auristatin E. (MMAE) with a cleavable linker for disruption of microtubule network	DLBCL–registration	PV was combined with RTX and bendamustine (pola-BR) and compared to bendamustine and RTX treatment alone. Pola-BR patients had a significantly higher CR rate (40.0% vs. 17.5%; *p* = 0.026) and longer PFS [176].
Brentuximab vedotin (BV)	CD30	auristatin E. (MMAE) with a cleavable linker for disruption of microtubule network	HL–registration ALCL-registration	75% and 86% OR rates in r/r HL and ALCL, consecutively [185].
Inotuzumab ozogamicin (CMC-544)	CD22	Calicheamicin with a cleavable (acid-labile) linker for disruption of double-stranded DNA in the nucleus	NHL B-ALL–registration	39% ORR in r/r/B-ALL (NCT01134575) CMC-544 in combination with RTX resulted in longer survival in a disseminated B-cell lymphoma murine model [186].
AGS67E	CD37	Monomethyl auristatin E. (MMAE) with protease-cleavable linker for disruption of microtubule network	NHL CLL	Until now only one clinical trial was completed, the safety of AGS67E was demonstrated in patients with r/r lymphoid malignancies (NCT02175433) [187].
Pinatuzumab vedotin	CD22	Monomethyl auristatin E. (MMAE) with a cleavable linker (with sulfhydryl groups) for disruption of microtubule network	NHL	Objective responses were observed in DLBCL (9/25); CR in 2/8 patients treated with pinatuzumab vedotin and RTX [188].
Naratuximab emtansine (IMGN529)	CD37	Emtansine (DM1) with a non-cleavable linker for disruption of microtubule polymerization	r/r B-cell lymphomas	In a phase 1 study on B -NHL, a reduction in lymphocyte count was observed after the second dosing of IMGN529 [189].
Denintuzumab mafodotin (SGN-CD19A)	CD19	Monomethyl auristatin F with a cleavable linker for disruption of tubulin polymerization	NHL	In a phase 1 study SGN-CD19A exerted durable responses in heavily pre-treated NHL patients; 56% objective responses were achieved for relapsed patients with a CR rate of 40% [190].
Coltuximab ravtansine (SAR3419)	CD19	Ravtansine (DM4) with a cleavable linker (with disulfide groups) for disruption of tubulin polymerization	DLBCL	In a phase 2 study on r/r/DLBCL, ORR was achieved in 43% of patients (18/41), where PFS and OS were 4.4 and 9.2 months, respectively [191].

### 6.4. CAR-T Cells

In recent years, considerable progress in hematooncology has been achieved by developing adoptive therapies based on T cells modified with chimeric antigen receptors (CARs) [192,193]. CARs targeting CD19 have been a breakthrough in the treatment of ALL and DLBCL. The results of clinical trials have been especially promising in ALL, where high CR (up to approx. 80%) was observed, while it was much lower (approx. 50%) in r/r DLBCL patients. Still, it has to be underlined that in r/r DLBCL prognosis is dismal; thus, each therapeutic option counts. The encouraging results of the clinical trials resulted in the registration of four different CAR CD19 formulations varying in the composition of the intracellular co-stimulation domain, three of which, namely: tisagenlecleucel, axicabtagene ciloleucel, and lisocabtagene maraleucel are approved for the treatment of DLBCL patients [193,194,195].

However, almost half of patients with high-grade lymphomas progress after CD19 CAR-T cell therapy and require additional treatment [196]. The mechanisms of tumor escape are related to host factors affecting CAR proliferation and persistence, features of CAR-T product and their toxicity, and tumor intrinsic factors, such as loss of target epitope expression or trogocytosis [192]. By now, the information on the mechanisms of resistance to CAR-based therapy comes mainly from observations in ALL.

In ALL, antigen loss is one of the most frequent reasons for relapse after CD19 CAR-T therapy. Disease relapse can occur by 12 months after infusion of CAR-T cells in up to 50% of patients [197]. Relapse connected with CD19 antigen loss is also observed in the case of NHLs but is less frequent than in ALL [198].

Thus far, two main mechanisms of antigen loss were described in ALL–antigen escape and lineage switch. In the first case, after remission in response to CD19 CAR-T therapy, patients lacking surface CD19 antigen relapse but with a phenotypically similar disease. In contrast, lineage switch describes the opposite situation, when relapse is a phenotypically different malignancy, but genetically similar [199]. That occurs because of several splicing variants of CD19 lacking the CAR-recognizing epitope [200,201]. It was demonstrated that relapsed ALL is characterized by a mutation in exon 2 of CD19 and the expression of other CD19 isoforms. This resistance mechanism supports BCP-ALL proliferation and prevents the killing by CD19 CAR-T [202]. Recently, the role of hypermethylation of CD19 promoter in antigen loss has been demonstrated in an in vivo CD19-negative recurrence model of CLL following CD19 CAR-T therapy [203]. Less frequently, the tumor escape is mediated by low antigen density or trogocytosis [192,204]. Moreover, as the effectors acquire CD19 expression at their surface, they are in consequence eliminated by fratricide killing [204]. Nevertheless, the recent study by Jain et al. has questioned the role of CD19 downregulation/loss in determining resistance to CD19 CAR-T in DLBCL and underlined the role of immune exhaustion [205]. By characterizing the genomic signature of DLBCL primary samples obtained from patients that subsequently underwent CD19 CAR-T administration, they discovered some tumor-intrinsic genomic alterations such as chromothripsis (aberrantly reassembled chromosomes leading to aneuploidies), the mutational activity of APOBEC, deletions of *RB1* or *RHOA* that according to the authors correlate with exhausted immune microenvironments and resistance/poor response to CAR therapy [205].

CD19 loss after CAR-T therapy in ALL can be accompanied by the downregulated expression of some surface antigens, such as CD20 and CD22 [206]. A similar effect of CD22 downregulation with CD19 loss after CD19 CAR-T treatment was observed in DLBCL patients [207]. Moreover, the CD58 aberration (lack of expression or mutation) is observed in some r/r DLBCL patients after CD19 CAR-T treatment. CD58, as a ligand for CD2 on T cells that provides the costimulatory signal, has been linked to resistance to CD19 CAR-T and progression after this therapy in DLBCL [208]. There is also some evidence that 10–20% of patients with ALL have primary resistance to CD19 CAR-T. This type of resistance can be correlated with lower expression of death receptors connected with apoptosis pathways in neoplastic cells as demonstrated in in vitro model testing CRISPR-based genome-wide loss-of-function screen in Nalm-6 ALL cell line [209]. The potential strategies against resistance to CAR T-cell therapy in hematological malignancies, including modifications in the structure of CARs, novel therapeutic targets, and safety-related issues have been elegantly reviewed in [192,193,210].

## 7. Perspectives

Hematological malignancies derive from immune cells and are surrounded by other components of the immune system. This provides many opportunities to interact with the tumor environment, locally transported antibodies, and the complement system. These characteristics shape unique conditions for the use of immunotherapy, which in the last two decades has become the mainstream for many types of hematological malignancies, including the most frequent one—DLBCL.

Despite extensive research on novel therapeutic options, R-CHOP combination remains the core of therapy in DLBCL. While the frequency of primary resistance to R-CHOP is relatively low (approx. 10%), secondary resistance acquired in the course of therapy leads to relapse in around one-third of patients [211]. The situation is particularly delicate in the case of r/r DLBCL patients, as their prognosis is dismal. In this population, the overall survival amounts to 6 months [5]. Therefore, r/r DLBCL patients urgently need novel therapeutic options. Given the genetic heterogeneity of this malignancy, targeting common B-specific antigens present on the cell surface with monoclonal antibodies or CAR-modified T- and NK-cells seems a rational option. Nevertheless, the years of employment of rituximab as a core of first-line treatment and recent observations on the mechanisms of resistance to CAR-T cell therapy suggest that treatment resistance is an inevitable consequence of applying immunotherapeutics. Therefore, a question arises: which therapeutic schemes will cause maximal efficacy against malignant cells in r/r patients? A thorough study of the mechanisms of primary and acquired resistance to immunotherapy may lead to optimization of the already existing compounds as demonstrated by the introduction of hexamerization-inducing mutations sensitizing to CDC and Fc-engineering of mAbs, leading to increased ADCC. Analyses of resistant cellular models and primary material from r/r patients can reveal immunotherapy-induced changes in the phenotype of malignant cells, which can be further addressed by designing specific mAbs and CAR-modified effector cells.

## Figures and Tables

**Figure 1 ijms-23-01501-f001:**
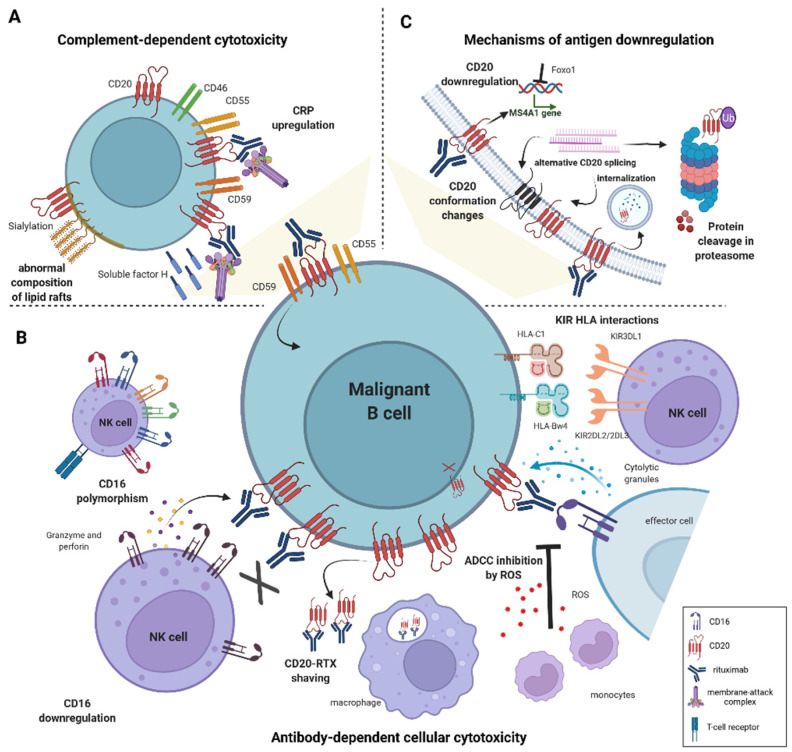
Mechanisms of resistance to anti-CD20-based immunotherapies in DLBCL. (**A**) Induction of resistance to complement-dependent cytotoxicity (CDC) is mediated by an increased expression of complement regulatory proteins (CRPs)–CD46, CD55, CD59 and soluble protective protein complement factor H (CFH). The abnormal composition of the lipid raft domains, e.g., hypersialylation compromises complement-induced cell lysis. (**B**) Impaired antibody-dependent cellular cytotoxicity (ADCC) is caused by genetic polymorphism in FcγRIII receptor (CD16) and low expression of CD16. Interactions between specific KIR ligands on NK cells and HLA molecules on B cells reduce NK-cell degranulation. Monocytes mediate shaving of RTX/CD20 complexes from the cell surface; macrophages and neutrophils reduce the anti-tumor potential of NK cells by releasing reactive oxygen species (ROS). (**C**) CD20 expression is regulated by numerous epigenetic and transcription factors, e.g., Foxo1. Deficiency in CD20 level arises from mutations in *MS4A1* gene leading to alternative splicing, and from CD20 internalization. Conformational changes in CD20 protein decrease binding affinity to anti-CD20 mAbs.

**Figure 2 ijms-23-01501-f002:**
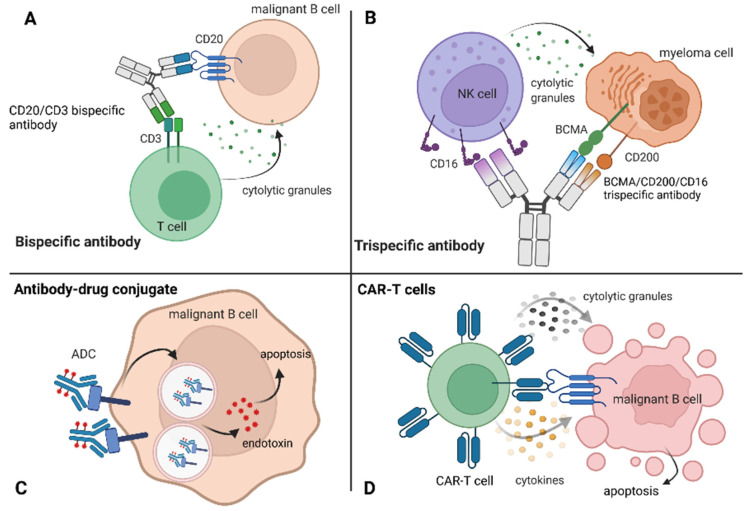
Targeted immunotherapies in hematological malignancies. (**A**) bispecific antibody; (**B**) trispecific antibody; (**C**) antibody–drug conjugate; (**D**) chimeric antigen receptor T cells.

**Table 1 ijms-23-01501-t001:** Selected bispecific antibodies in clinical and preclinical studies.

Name	Target	Type	Mode of Action	Efficacy in Clinical and Preclinical Studies
Blinatumomab	CD3/CD19	bispecific antibody (BiTE)	CD3-positive T-cell recruitment	In phase 2 study on r/r DLBCL patients (NCT01741792), who received 3 prior lines of therapy blinatumomab induced OR in 43% of patients (9/21), including CR in 19% of subjects (4/21) [159]. In heavily pre-treated r/r B-NHL patients (*n* = 73) receiving blinatumomab long-term remission (over 7 years) was achieved in a subset of patients [160,161].
Epcoritamab (DuoBody-CD3xCD20)	CD3/CD20	bispecific antibody (BiTE)	CD3-positive T-cell recruitment	Epcoritamab induced potent cytotoxicity against primary B-NHL samples (DLBCL, FL, MCL) irrespective of whether the patients were newly diagnosed or were r/r after CD20-mAb treatment [162,163]. In phase 1/2 clinical trial on 68 r/r B-NHL patients (NCT03625037) epcoritamab showed ORR in 86% of patients, including CR in 45% of subjects with no grade 3 or higher adverse events [164].
[(CD20)2xCD16]	CD20/CD16	bispecific tribody	CD16-positive effector cell recruitment	In preclinical studies ((CD20)2xCD16) demonstrated a 9-fold enhancement in ADCC and promoted 10-fold higher NK-cell number activation in comparison to RTX, irrespective of *FcγRIIIa* polymorphism [165].
NI-1701	CD19/CD47	bispecific antibody	ADCP enhancement via blocking immune checkpoint receptor	Blocking CD47 with mAb increased phagocytosis of NHL cells in preclinical studies [166] and in a small clinical study in DLBCL patients (NCT02953509). NI-1701 induced potent cytotoxicity against B-lymphoma and leukemic cell lines in vitro and in NOD/SCID mouse xenograft model. It was more effective than monovalent anti-CD19 and anti-CD47 mAbs and achieved a synergistic effect with RTX [167].
CD20-HLA-DR DVD-Ig	CD20/HLA-DR	bispecific antibody	Increase in selectivity against NHL cells versus healthy cells	CD20-HLA-DR DVD-Ig demonstrated potent CDC and ADCC against B-cell lymphoma in vitro by inducing homotypic adhesion and actin reorganization. It effectively depleted Raji cells from a mixture with whole human blood and showed high specificity towards malignant cells, leaving normal B cells unaffected [168].
Bs20×22	CD20/CD22	bispecific antibody (RTX/HB22.7 platform)	Induction of apoptosis	Bs20x22 induced a 3-fold higher rate of apoptosis than parent mAbs (anti-CD20 RTX and anti-CD22 HB22.7) despite unchanged binding affinity. It also increased survival in nude mice Raji xenograft model (88%) as compared to other treatment regimens (RTX and Bs20x22–75%, RTX-50%, HB22.7–25%) [169].
Bs20x22	CD20/CD22	bispecific antibody (hA20 (veltuzumab)/hLL2 (epratuzumab) platform)	ADCC enhancement via antigen crosslinking	Tetravalent anti-CD20/22 bsAb constructs demonstrated remarkable redistribution of CD20, CD22, and BCR into lipid rafts following cross-linking of both antigens leading to the initiation of phosphorylation cascade and internalization of Ab-BCR complexes [170].

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
