# Peer review of "Molecular Aspects of Resistance to Immunotherapies—Advances in Understanding and Management of Diffuse Large B-Cell Lymphoma"

_ijms, 2022, doi:10.3390/ijms23031501_

Round 1

Reviewer 1 Report

The present manuscript provides an extensive review on the currently known mechanisms of resistance to immunotherapy in DLBCL. It is well written and organized. However it is extremely long and not easy to read. I would suggest that a significant reduction should be made in all the sections of the manuscript. 

Author Response

We would like to thank the Reviewer for the time and consideration. According to the Reviewer’s suggestions, we have carefully examined the entire manuscript and we have reduced the content of all the sections to the highest possible extent. Specifically, we introduced a significant reduction in the sections: 1. Introduction, 3.4. Hexamerization-inducing mutations as a way to improve CDC efficacy, 6.2. Bispecific antibodies, and 6.3. Antibody-drug conjugates, preserving consistency of the manuscript. We hope that the manuscript has been improved towards the required standards after this revision and will be
more attractive and easy to read for the readers.

Reviewer 2 Report

The authors report molecular aspects of resistance to immunotherapies-advances in understanding and management of diffuse large B-cell lymphoma.

  1. The authors should add the name of receptors in the Figures. Also, the authors should add the summary data of intracellular signaling in a Figure. It will be benefit for the reader.

Author Response

We would like to thank the Reviewer for this important comment. Thanks to the Reviewer’s suggestion, we have added a legend describing receptors depicted in Figure 1 and included their names in the main body of the Figure. Moreover, to facilitate the interpretation of Figure 1, we have introduced a summary explaining the most important molecular mechanisms and interactions associated with the various form of resistance to anti-CD20 based immunotherapies. Finally, we added the missing names of antigens and receptors in Figure 2. We hope that these changes will improve understanding and interpretations of both Figures.

Reviewer 3 Report

The review is very well-written and well-organized. It extensively describes and discusses different molecular aspects that influence the success of immunotherapiutic approaches in the treatment of DLBCL and  of the development of resistance. The authors summerized the available literature about the topic interestingly highligthing wich kind of experimental models are yet missed.

In my opinion this manuscript is acceptable for publication in the present form.

Author Response

We appreciate the encouraging comment of the Reviewer. We are pleased that the Reviewer considers our manuscript suitable for publication in the present form and we hope that the readers will find it valuable as well.

Round 2

Reviewer 2 Report

none